# Robotised Geometric Inspection of Thin-Walled Aerospace Casings

**DOI:** 10.3390/s22093457

**Published:** 2022-05-01

**Authors:** Artur Ornat, Marek Uliasz, Grzegorz Bomba, Andrzej Burghardt, Krzysztof Kurc, Dariusz Szybicki

**Affiliations:** 1Pratt and Whitney Rzeszów S.A., 35-078 Rzeszów, Poland; marek.uliasz@prattwhitney.com (M.U.); grzegorz.bomba@prattwhitney.com (G.B.); 2Department of Applied Mechanics and Robotics, Faculty of Mechanical Engineering and Aeronautics, Rzeszów University of Technology, 35-959 Rzeszów, Poland; andrzejb@prz.edu.pl (A.B.); kkurc@prz.edu.pl (K.K.); dszybicki@prz.edu.pl (D.S.)

**Keywords:** robot, programming, laser measurement, database, robotised process

## Abstract

This paper deals with the development of dimensional control technology for the production of ADT (accessory drive train) gearbox housing. The project included the development of a robotic geometry inspection station for thin-walled aerospace casings. Laser profilometers from two brands, Cognex and Keyence, were used to measure. A proprietary software solution for arc measurements is presented. The obtained solutions were compared and verified in accordance with the requirements of the manufacturer Pratt and Whitney Rzeszów S.A. The results of the work indicated that correct solutions were obtained with a very large reduction in control time. In addition, the measurement is fully automated and transferable to the company’s electronic systems.

## 1. Introduction

During the production and quality inspection of thin-walled aerospace casings, a great number of geometric parameters must be measured. The parameters can be acquired with three different methods: by taking imprints and displaying their values with a measurement projector; a contact method using a hand-operated laser profile measurement tool; or a contact method using a contact probe operated by a CMM (coordinate measuring machine). Each of these methods has advantages and weaknesses.

When taking imprints, its slicing and display with a measurement projector is a time-consuming process, burdened with the potential for measurement errors as it is directly handled by a human (human factor). A manual measurement method with a hand-operated laser profile measurement tool does not ensure a reliable result given the potential incorrect orientation of the tool’s head, which should be perpendicular to a tangent of the measured surface, difficult access to the measured area, measurement operator’s fatigue, etc. The first of the two methods discussed requires considerable time, is prone to human factors and does not facilitate the automatic generation of measurement reports.

Measurements with CMMs can only be undertaken in suitable laboratory conditions. This method requires the removal of the casing to be measured from the machine fixture, handling the casing to a testing laboratory, waiting until the casing reaches temperature equilibrium, and installing the casing with the reference systems retained inside of the measuring area of the CMM. This requires more time and space. The advantages of CMMs are the great flexibility of applications and accuracy. The greatest weakness of this method is the high purchase and maintenance cost of a CMM and the long measurement time [1,2].

Minimizing production costs implies, among other things, reducing time-consuming product inspection processes, which results in the search for alternative methods. Operations of dimensional and geometric tolerance control have so far often been performed on coordinate measuring machines (CMMs). Contact measurement with a CMM has well-defined accuracy and measurement uncertainty [3]. A cheaper alternative is non-contact measurements, unfortunately, the measurement uncertainty of such solutions is at least one order of magnitude higher [4]. Non-contact measurements, however, have an advantage in the speed of measurement data acquisition, allowing a higher density of points and a rapid acquisition of cloud points. Therefore, they have been widely used in reverse engineering [5,6]. In the last few years, a great effort has been made to improve the accuracy of laser systems [7], and their use in inspection activities is increasing every day. The most common laser systems used in metrology applications are those based on laser triangulation using a laser beam (ruler). The reason for the popularity of 2D scanning technology is the relatively low price, high precision, and relatively low computational cost compared to methods based on structured light, holography, or image analysis. Interesting information can be found in work [8], where two different scanning systems were analyzed: a laser triangulation sensor and a contact probe, both mounted on a coordinate measuring machine, and the results were analyzed to compare them in terms of geometric and dimensional tolerances.

Measuring geometrical features such as full circle radii, chamfer widths, small diameters, or distances with a 2D scanner is a typical application solution and does not present a problem. The main problem solved in this paper is to give an algorithm for measuring radii performed as blunts of sharp surfaces. The difficulty arises from the need to adapt the measurement field, the number of selected points, and to propose a way to average the measurement. This is an approximation problem of inscribing a radius into a point-defined curve, realized under the software limitations of the Keyence and Cognex 2D scanner controllers being used.

In the era of automating industrial solutions, a robotised geometric inspection of workpieces was proposed; the system is based on laser beam measurements with profilometers manufactured by Cognex and Keyence, whose displacement, positioning, and orientation are performed by a robotic arm. The specific nature of geometric measurements required the development of processing algorithms for the results and the design of a robotised inspection station. A verification measurement could be triggered during the machining of the casing or directly after the machining process, with the measurement results stored in a database to retrieve a final inspection report.

A Pratt & Whitney-approved measurement method using imprints was used as the standard to which the obtained measurement results were related.

## 2. Measurement Methods Applied

An example of thin-walled aerospace casings is the ADT (accessory drive train) gearbox. The ADT (Figure 1) is a jet engine component that transfers power to the engine accessory systems: lubrication, ventilation, fuel delivery, control, electrical, and hydraulic.

Each transmission consists of several hundred components, the most important being the main body gearbox and cover. The gearbox must meet the highest requirements in terms of quality and precision, which is why it is essential to check the geometric parameters during production and at the release stage. The production of ADT gearboxes for the P&W turbofan engine PW1500G series used in Airbus A220 aircraft involves manufacturing thin-walled aluminium alloy castings with a complex geometry [1,2]. Such requirements are dictated by the need to achieve the highest possible power density in relation to the weight of the transmission, which affects the cost-effectiveness of manufacturing the entire drive unit. The ADT is subject to certification as-assembled with its engine pursuant to EASA (European Aviation Safety Agency) CS-E regulations, designed to ensure the operating safety of aircraft equipment. This determines the process of continuous quality control of the performed technological operations.

For aircraft engine components, such as the ADT gearbox, each manufactured ADT is inspected. There are more than 1000 dimensional and shape features (the measurements of various geometric dimensions) produced in various steps of the manufacturing process for each ADT gearbox. It should be noted that the places where measurements need to be taken are often difficult to access. It is because these ADT gearboxes are designed for the installation and interposition of various parts, assemblies, and mechanisms, which mainly include bearings, shafts, and gears. It is achieved with the minimum possible weight of the ADT gearbox and limited installation space. The limited installation space results in a complex ADT gearbox shape. It is also a consequence of the fact that the external shape of the casings is not much different from the contours of the internal components.

As already mentioned, the measurement of geometric features is often conducted with CMMs. Note that not every measurement requires the level of accuracy which demands the application of an CMM (Figure 2).

Local features, such as fillet radii or chamfering width are measured using imprints (Figure 3a) and a measurement projector (Figure 3b).

This type of measurement requires the imprint of the geometric feature to be manually taken; the imprint is sliced and then measured by an operator using a measurement projector.

The local parameters of the casing can also be measured with a hand-held laser measurement tool called GapGun Pro (Figure 4).

Both these manual methods are outdated, burdened with the weaknesses discussed before, and unsuitable for the challenges of modern industry. In the course of this work, a robotised system was proposed for the measurement of local geometric features, which are the majority of the 1000 features performed on the ADT casing.

To perform a robotised geometric measurement correctly, a suitable measurement device is required. The measurement device can be stationary, for instance, and designed and built with ready-made measurement modules [9]. Market reconnaissance and a review of modern measurement devices feasible for implementation on a robotic arm pointed to the following: Leica Absolute Tracker, Leica Absolute Scanner, MetraSCAN 3D R-Series, Gom Atos Core, Cognex DS910B, and Keyence LJ-V7200 or LJ-V7060.

Leica Absolute Tracker is a portable all-in-one laser tracker device. The AT960 tracker is a comprehensive 6DoF solution (the abbreviation means six degrees of freedom), designed for probing, scanning, automatic inspection, and reflective measurements. It can be complemented with the Leica Absolute Scanner LAS-20-8 for 3D discretisation. An application example of such sa canner in robotised measurements is published in [10], where the authors applied the tracker to measure the position and orientation of the TCP (tool centre point) on a robot. With a real-time user interface, the authors acquired the positional data in millisecond cycles, and the data were used to calculate the actual errors of the robot’s trajectory of motion. In [11], a result of the integration of a real-time laser tracker with a milling robot is presented. It is a high-performance alternative to automated machining of large components in the aerospace industry. The integration of an external control loop enabled the improvement of the robot’s positioning accuracy. According to the information provided by the system’s vendor, its costs had greatly exceeded the available funding, and the solution was abandoned.

The MetraSCAN 3D R-Series was showcased in [12] as a next-generation solution because manufacturing companies will often inspect all manufactured parts before shipping. Customers will require access to the inspection data before accepting the delivery of ordered parts. This need for the traceability of each manufactured part requires a quality system which facilitates more inspections per hour to track the production pace while optimising the manufacturing process. The MetraSCAN 3D R-Series solution also proved to be very expensive. The considerable cost of the system precluded its application in the robotised inspection station.

An application of the Gom Atos Core with the Atos Professional software was presented in [13,14,15] as a measurement device in a robotised initial inspection of a thin-walled stator casting process. If the casting meets the assumed tolerances, the characteristic coordinates of the casting points are determined for further adaptation during alternative robotic measurement of the blade airfoil wall thickness using the UTT method in 168 points. The paper [16] presents a proprietary method for data communication and exchange in a robotic machining cell which comprises two robotic manipulators, a positioner, and a 3D optical scanner. One of the robots, tooled with the 3D optical scanner, was tasked with acquiring the cloud of points of a workpiece fixed on the positioner table to detect allowances. The coordinates of the allowances, which were stored in a script, were sent to the controller of the second robotic manipulator for downstream processing, adjusting the parameters to their random values. This solution, however, was done without due to the costs and the time of scanning and polygoning. Moreover, it was recommended to superimpose an anti-reflective coating on the workpiece being measured to eliminate the reflection of light, which are unacceptable during fast robotic measurements.

The Keyence LJ-V7200 was presented in [17] as a device improving the accuracy of assembly robots in the aerospace industry. An online 3D positioning system was proposed, comprising the Keyence LJ-V7200 scanner and a 6Dof tracking system (built with the Leica AT-960 and T-Mac TMC-30B). The average 3D positioning accuracy for the object in the workspace of the LBR-iiwa robot was ± 0.16 mm. In the paper [18], this measurement head assembly was applied in the scanning of HPGR cylinders to detect their wear. The feasibility of continuous scanning of the cylinders provided much information about all deviations from the design values and their effect on the wear pattern. The Keyence LJ-V7200 also found other applications [19,20,21,22] as a stationary device for the detection of component wear or out of specification instances.

Industrial robots are used for a wide variety of operations, from simple moving materials from point A to point B to complex operations such as metalworking or assembly using sensory systems to adapt to a randomly changing environment [23,24]. Another example of adaptation algorithms or robot path correction based on camera image analysis in a welding application is included in [25].

A literature review did not reveal any applications with the Cognex DS910B [26] laser head in a solution forming a measurement device of robotic systems, much less on a robotic arm. Together with the Keyence LJ-V7060 [27], these solutions are smaller in size, provide faster measurements, and are less expensive than the previously discussed ones. The data referenced herein prompted the authors hereof to test the feasibility of the solution in geometric inspection measurements conducted with an ABB IRB 2400 robot. The measurement heads qualified for the test required building a communication system for the robot’s controller and developing the measurement software.

The developed measurement system is intended to replace geometric measurements with imprints and complement CMM tests. The measurement system’s accuracy should be comparable to that of the imprints method. The difference between the measurements made with imprints and the planned laser measurement system shall not exceed 0.05 mm.

## 3. Proposed Solution

The system was built with the ABB IRB 2400 industrial robot and its IRC5 controller. The robot’s repeatability guaranteed by the manufacturer was 0.03 mm, with an accuracy lower by an order of magnitude. It did not affect the capabilities of the inspection system, as only the local features were measured, such as the chamfer width and fillet radius. The task of the robot was to move the measurement head to pre-programmed measurement locations. The measurement accuracy was driven by a 2D scanner and its software. It is important that even the robot’s repeatability did not affect the measurement error, as the task of the 2D scanner software was to determine the measurement area. Each measurement is made locally; the accuracy of the measurement depends only on the accuracy of the measuring head being used, defined by the manufacturer. The manufacturer states that the Cognex DS910B provides a measurement accuracy of 7 μm, while the Keyence LJ-V7060 provides a measurement accuracy of 5 μm. The measurement system consisting of Kyence and Cognex 2D laser scanners is calibrated by the manufacturers, and the accuracy of their measurements is defined in the manufacturer’s certificate.

Other components of the inspection workstation included an ABB IRBP A250 2-axis positioner, on which the workpiece and its fixture were mounted (Figure 5). The workpiece is mounted on a positioner relative to the mounting bases. The positioner’s role is to position the workpiece properly for measurement. The ABB IRB 2400 robot (with six degrees of freedom) on its own is not capable of providing proper head position and orientation relative to the workpiece due to the size of the measured workpiece. The repeatability of the positioner does not affect the accuracy of the measurement because the measurement is local to the indicated location and depends only on the accuracy of the measuring head used.

In the course of this investigative work, two solutions of laser scanners were tested—the Cognex DS910B and the Keyence LJ-V7060. The scanners communicated over TCP/IP and Ethernet (Figure 6).

The operating algorithm of the robotised inspection station was developed in robot ABB language RAPID and the communication was based on TCP/IP (Figure 7). The data communication was between the robot’s controller and the PC with the Cognex Designer software (which acted as the controller for the scanner) or the Keyence XG-X2800 controller. For both controllers, the inspection measurement software was developed with the type of measurement. The inspection measurement software required adaptation to the location and the type of measurement, such as the chamfer or radius measurement.

The robot’s controller program comprised variables and commands for the movement of the tooling (the 2D scanner) and instructions for the data communication with the scanner controllers. Communication with the scanners is bidirectional, with the robot reporting when a point has been reached and the characteristic number (the shape being measured). The feature’s sequential number served as the unique identification of the feature. The scanner controllers make the measurement once they receive the ‘arrival at point’ input (Figure 8). The feature’s sequential number identified the type of measurement (chamfer width or fillet radius, respectively) and the width of the measurement area.

The measurement end feedback and the measured value were sent to the robot’s controller. The robot’s controller saved the measurement to a database, with a comparison of the actual (measured value) to its nominal value and determined the deviation.

In the course of work, the capabilities of both tested inspection systems were compared. This paper focuses on the fillet radius measurement given its specific nature, a greater difficulty, and the need to develop advanced algorithms for the performance of this type of measurement.

## 4. Verification of the Proposed Solution

ADT’s dedicated test housing has more than a thousand dimensional characteristics such as angle, radius, distance, and measurements of the relationship between bearing seats such as concentricity, flatness, etc. The proposed measurement method applies only to measurements of local features with dimensions dependent on the width of 2D scanner measurement lines. For gearboxes, these types of measurements account for 83% of the total measurements made. A number of the controlled measurements are defined as critical dimensions, and in this case, the measurements are mainly performed on CMM measuring centers. Of all the critical dimensions, only 82 measurements are made using a 2D scanner. The proposed solution does not eliminate all the necessary measurements, but it significantly speeds up the inspection process; this applies to the accuracy of chamfer and radius blunts as well as to the measurement of small diameters.

The measurements of fillet radii are critical in ADT casings, as their quality of manufacturing highly determines the strength of the ADT. The measurement was localised, as it was required in several hundred locations of the ADT, urgently prompting automation of the process. It is important that for the ADT as the workpiece selected for the tests, the range of measured fillet radii was within 0.4–1.8 mm, and the nominal fillet radius value varied with the measuring location (Figure 9).

Both tested inspection systems functioned alike. The algorithm applied to determine the fillet radius is shown in Figure 10.

When the scanner head was brought to the predefined measurement location, the robot output the feedback releasing the measurement. An image of the geometric profile was acquired by the CCD matrix of the scanner (Figure 11).

In the next step of the process, the inspection system used suitable filters to convert the acquired image into a curve for downstream processing (Figure 12).

To automatically determine the centre of the measurement area (the rectangle), the arms were approximated with a straight line (Figure 13) and the intersection was determined (Figure 14a). A circle was inscribed in the rectangle, with the radius being the sought value (Figure 14b).

Figure 11, Figure 12, Figure 13 and Figure 14 illustrate the measurement of a fillet radius with the application of the solution developed by Cognex; for the Keyence solution, the procedure was similar (Figure 15).

The software for both tested solutions determined the fillet radii and output the measured value to the robot’s controller. During the measurements of the fillet radii limited to 0.4–1.8 mm in size, it became evident that the measured value depended on the measurement area’s width for both tested systems. As this dependency prevented automatic geometric measurements, a root cause analysis was undertaken in an attempt to solve the problem.

The measurement area (the rectangle) was the area of curve calculation (Figure 16). The inspection software operated by approximation or fitting the curve to the determined measurement locations (points), where the curve did not precisely cross the points but reflected a general data trend.

For the vision system, the search of the curve that crossed each measurement point would be incorrect and lead to false conclusions in the analysis. The model applied to approximate the measurement data can have different forms; however, the form used most often is low-degree polynomials. For the fillet radius inspection, a 2nd-degree polynomial approximation is used, an equation of a circle [28,29].
(x − A)^2^ + (z − B)^2^ = r^2^,(1)
with: x, z—the coordinates limiting the motion of the scanners; A, B—the coordinates of the circle centre.

Depending on how the approximation error was measured, two approximation types exist:steady approximation;mean-square approximation.

For a steady approximation, it is assumed that functions z = f(x) and z = F(x) are determinate and continuous in the interval of [x1; x2]. The approximation error is measured with the Chebyshev standard [30].
‖f − F‖ = sup_x1 ≤ x ≤ x2_‖f(x) − F(x)‖.(2)

For a mean-square approximation, two cases may exist:continuous approximation, where function z = f(x) is determinate in the interval of [x1; x2]. The approximation error is expressed with the following relation:
(3)$$\|f\;-\;F\|\;=\;\overset{x2}{\underset{x1}{\int }}w(x)[f(x)\;-\;F(x){]}^{2}\;\mathrm{dx},$$
with z = w(x) being a non-negative, real weight function

discrete approximation, where function z = f(x) is discrete, meaning that its known values can be tabulated.

For the inspection software discussed here, a continuous mean-square approximation was performed. It was not possible, however, to intervene in the method of approximation or to determine the approximation error; controller software developers do not provide this option. The only accessible, programmable parameter was the measurement area width, or the interval of [x1; x2]. To determine the relation between the measurement area width and the measurement accuracy, it was decided to measure the gauge elements made on the CNC machine with the accuracy of 0.05 mm, and this value was taken as the nominal one in the tests (Figure 17). In this paper, the accuracy of the proposed solution was checked on a reference part approved at the gearbox manufacturer’s plant (Figure 17).

During the tests, for known fillet radii the measurement area width was increased in increments of 0.1 mm (Figure 18).

The starting width of the measurement area was 0.3 mm. It was found that the measured fillet radius value was below the nominal value, followed by an increase and stabilisation around the nominal value in several successive steps of digitisation. When the limit value of the measurement area width was exceeded, the measured value greatly exceeded the nominal values. The tests were completed for a range of fillet radii from 0.25 mm to 3.5 mm. The completed tests facilitated the determination of the relation between measurement area width and inspection system error and the development of a measurement algorithm that could be implemented with the available hardware (Figure 19).

The algorithm (Figure 19) was successfully implemented both in the Keyence controller and the Cognex Design 4.3 software [31]. A part of the algorithm implemented in the Keyence controller is shown in Figure 20. Figure 20 is an image of the fragment as seen on the controller with the software implemented.

In the course of the tests of the developed inspection system solutions, the functioning of the proposed algorithms was verified.

## 5. Discussion

A comparison was made between the results produced with the algorithms developed for the Keyence and Cognex solutions with the method of taking an imprint by a qualified employee of Pratt and Whitney Rzeszów S.A. Table 1, Table 2 and Table 3 list the measurement results for three different fillet radius values.

To test the functioning of the inspection systems, fillet radii R01, R02, and R03 were measured in the assumed range. Each fillet radius was measured five times. The reference value adopted was the imprint measurement method, as this method was validated and approved by Pratt & Whitney S.A. Arithmetic mean was determined from the measurement values, and standard deviation was determined for the measurements made with laser profile measurement systems. The produced results proved to be satisfactory, according to prior assumptions.

## 6. Conclusions

For this paper, a robotised geometric inspection of thin-walled aerospace casings was developed and characterised. Existing methods of geometric feature measurement were analysed, and their advantages and weakness were listed. Automatic measurement methods viable for application with industrial robots were reviewed. The measurements with laser profile measurement systems were selected, and two solutions for these measurements, each provided by a different manufacturer, were tested. The main contribution of the research is the development of the algorithm and its implementation in the measuring device. This algorithm is designed to increase or decrease the width of the measuring field depending on the tested radius (0.4 to 1.8 mm) and to determine its different values automatically. The developed geometry inspection systems, together with the implemented algorithms, successfully performed the designed test tasks. The robot’s parameters, such as accuracy and repeatability, had no effect on the measurement accuracy. The test results achieved were satisfactory; following consultation with the representatives of Pratt and Whitney Rzeszów S.A., the discussed geometric inspection system solution can be potentially applied in a production cell. The robot ensured satisfactory repeatability of positioning and orientation relative to the workpiece measured by the scanner heads discussed here, which, serving as geometric measurement tools, proved to be very precise and fast in operation (with approximately 3 s for a single measurement point). Following the measurement inspection, the geometric values acquired were stored in an electronic database for post-processing. What is also especially important, the measurements could be integrated with the ADT casing machining workstation. In the process of machining, check inspections of specific parameters can be performed and correct the features that these parameters are (i.e., to rework fillet radii, chamfers, and more).

## Figures and Tables

**Figure 1 sensors-22-03457-f001:**
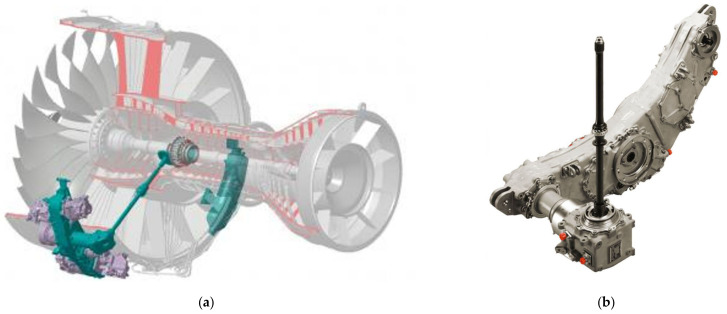
ADT: (**a**) Installation location on an engine; (**b**) actual component overview.

**Figure 2 sensors-22-03457-f002:**
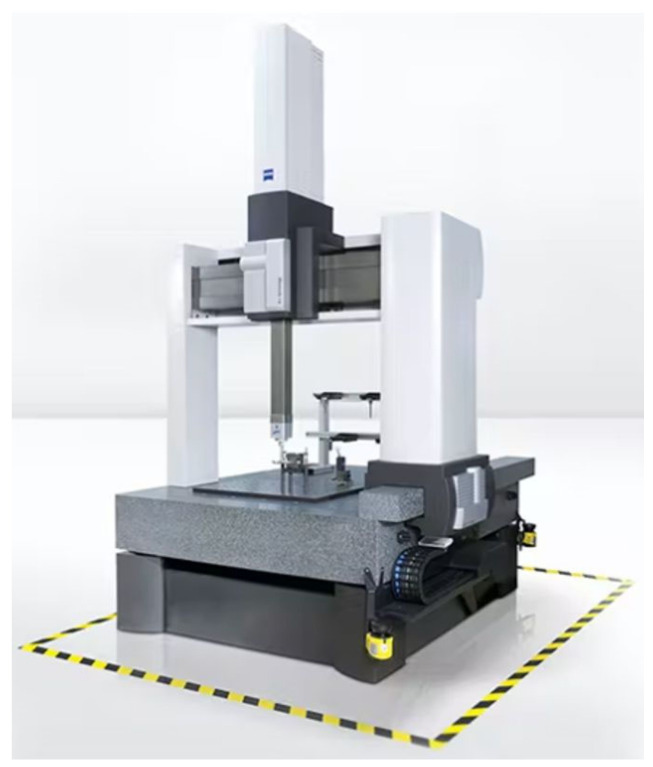
CMM (Coordinate Measuring Machine).

**Figure 3 sensors-22-03457-f003:**
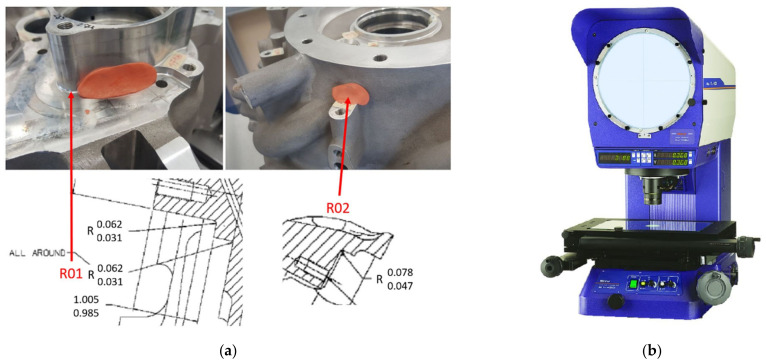
Measuring geometry using imprints, (**a**) taking imprints, (**b**) measurement projector.

**Figure 4 sensors-22-03457-f004:**
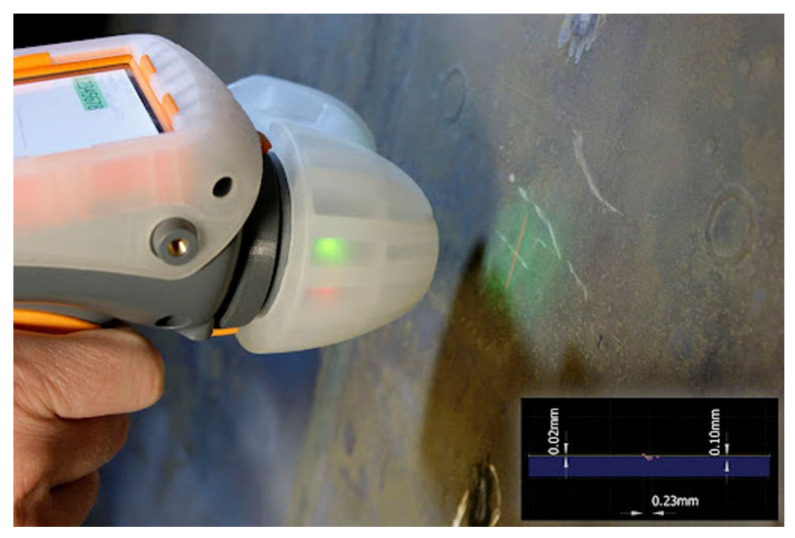
GapGun Pro hand-held laser measurement tool.

**Figure 5 sensors-22-03457-f005:**
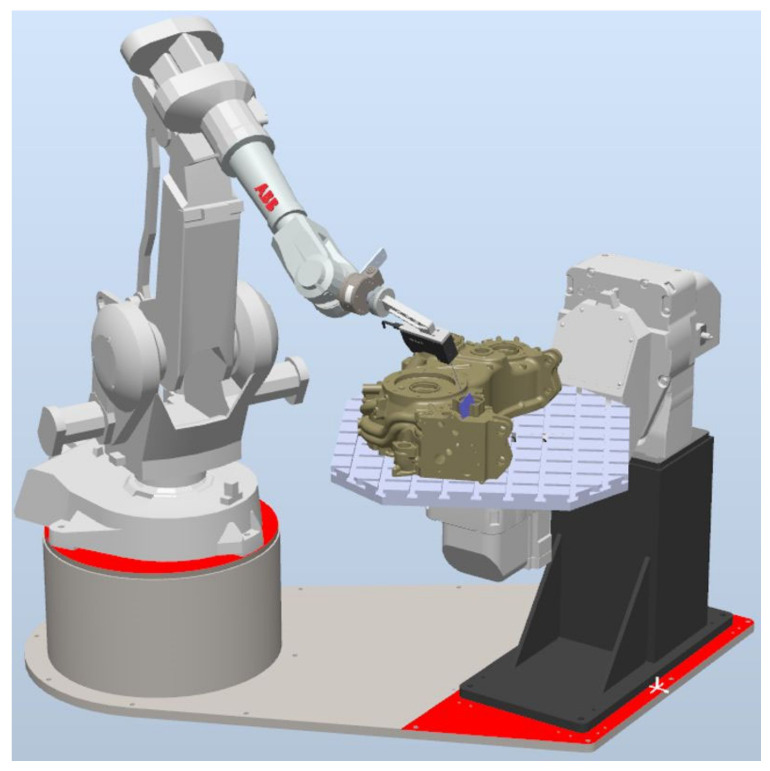
Inspection workstation’s model in RobotStudio.

**Figure 6 sensors-22-03457-f006:**
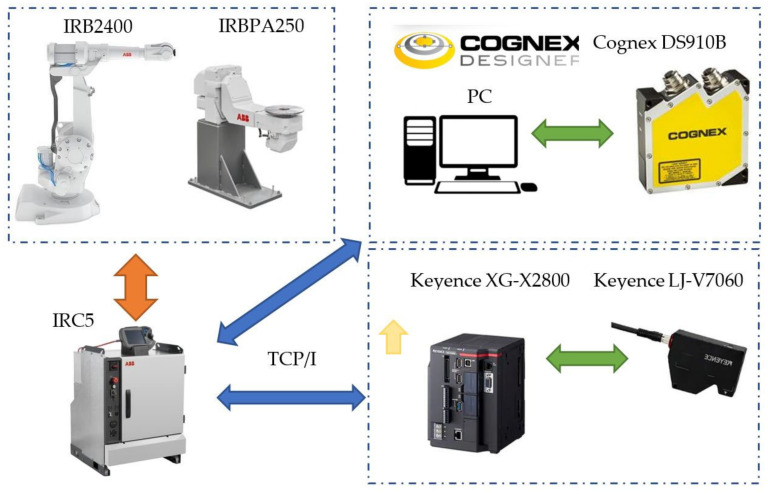
Data communication between the robot’s controller and the Cognex software.

**Figure 7 sensors-22-03457-f007:**
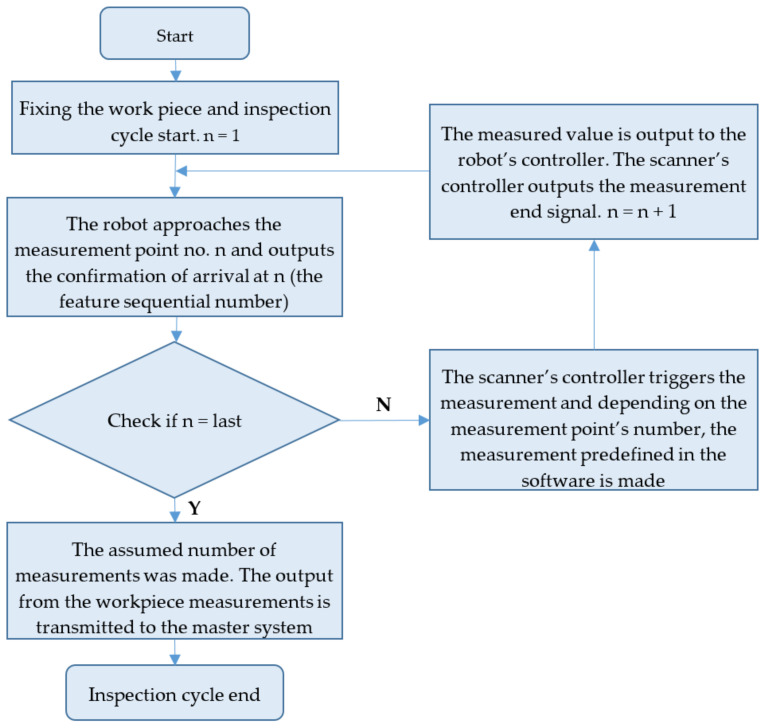
Operating algorithm of the robotised inspection station.

**Figure 8 sensors-22-03457-f008:**
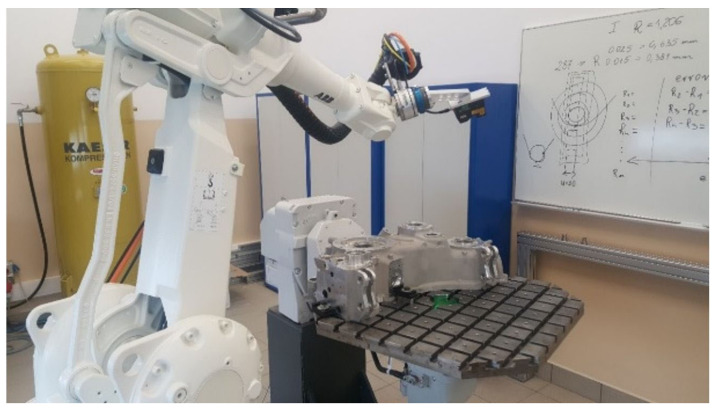
Robotised measurement.

**Figure 9 sensors-22-03457-f009:**
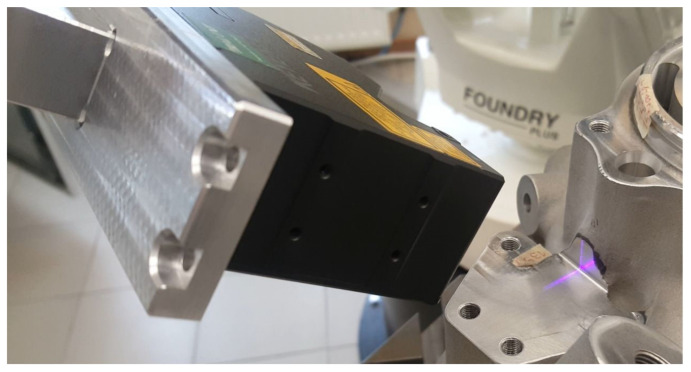
Robotised fillet radius inspection in one of the locations.

**Figure 10 sensors-22-03457-f010:**
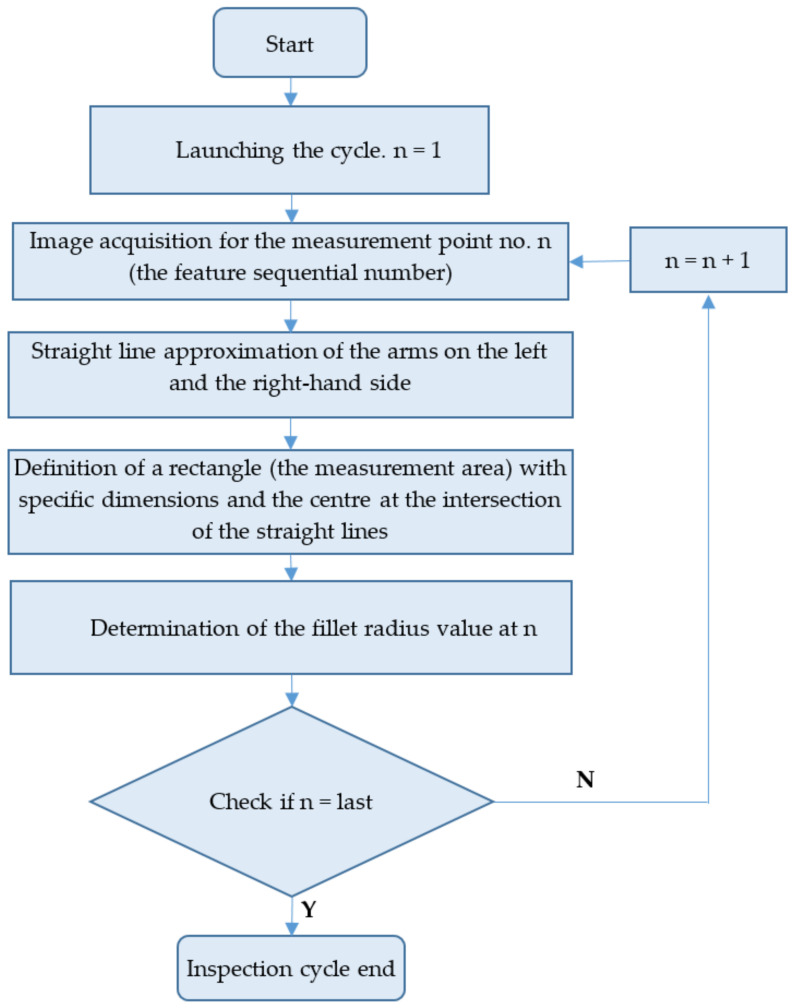
Fillet radius determination algorithm.

**Figure 11 sensors-22-03457-f011:**
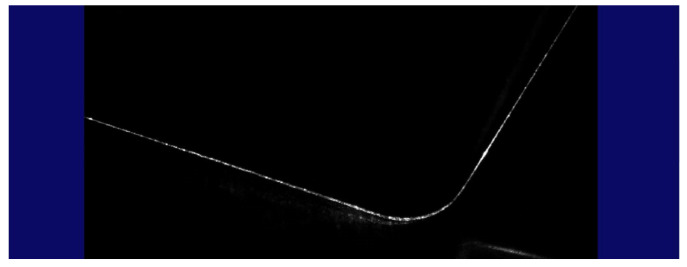
The image was recorded by the inspection system.

**Figure 12 sensors-22-03457-f012:**
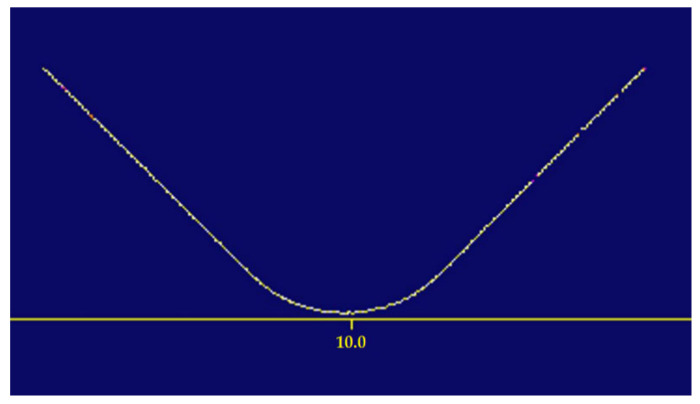
Resultant curve.

**Figure 13 sensors-22-03457-f013:**
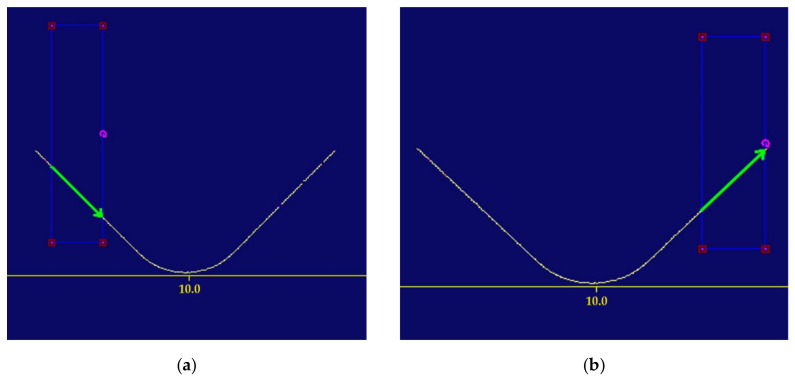
Straight line approximation of the arms on the left (**a**) and the right-hand (**b**) side.

**Figure 14 sensors-22-03457-f014:**
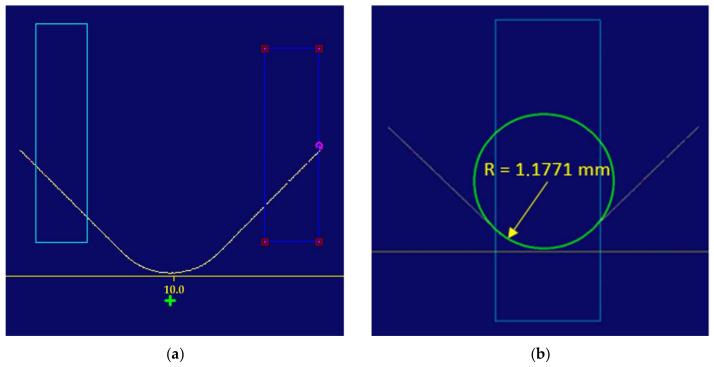
(**a**) Determination of the intersection point of the straight lines produced by approximation; (**b**) The measurement area rectangle and the determined radius.

**Figure 15 sensors-22-03457-f015:**
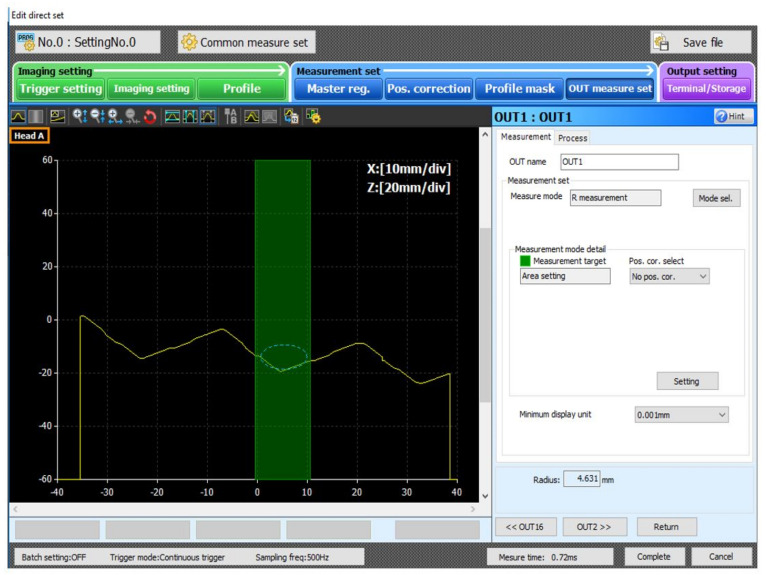
Definition of the measurement area rectangle and the determination of the fillet radius value in the Keyence software.

**Figure 16 sensors-22-03457-f016:**
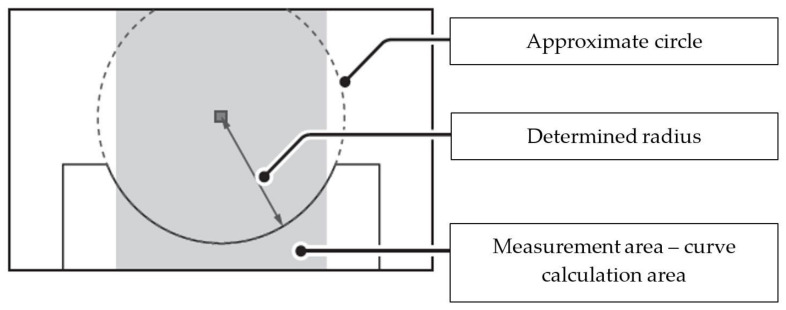
The concept of radius determination in the predefined measurement area.

**Figure 17 sensors-22-03457-f017:**
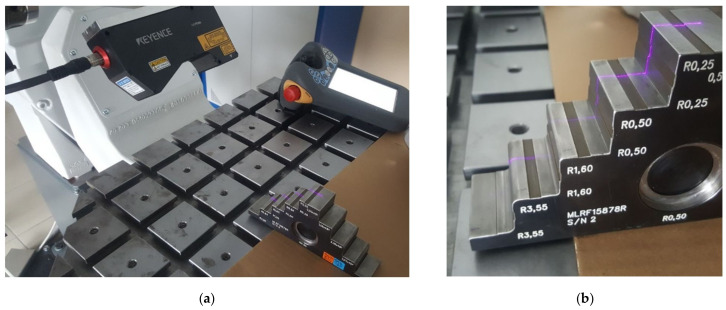
Measurement of the gauge elements: (**a**) overview; (**b**) detailed view of the gauge elements.

**Figure 18 sensors-22-03457-f018:**
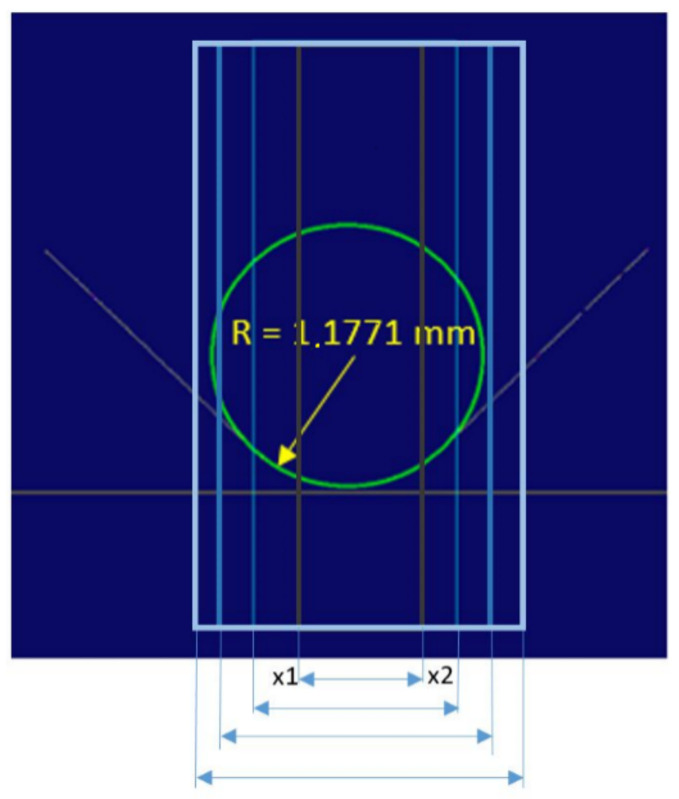
The concept of measurement area width increase.

**Figure 19 sensors-22-03457-f019:**
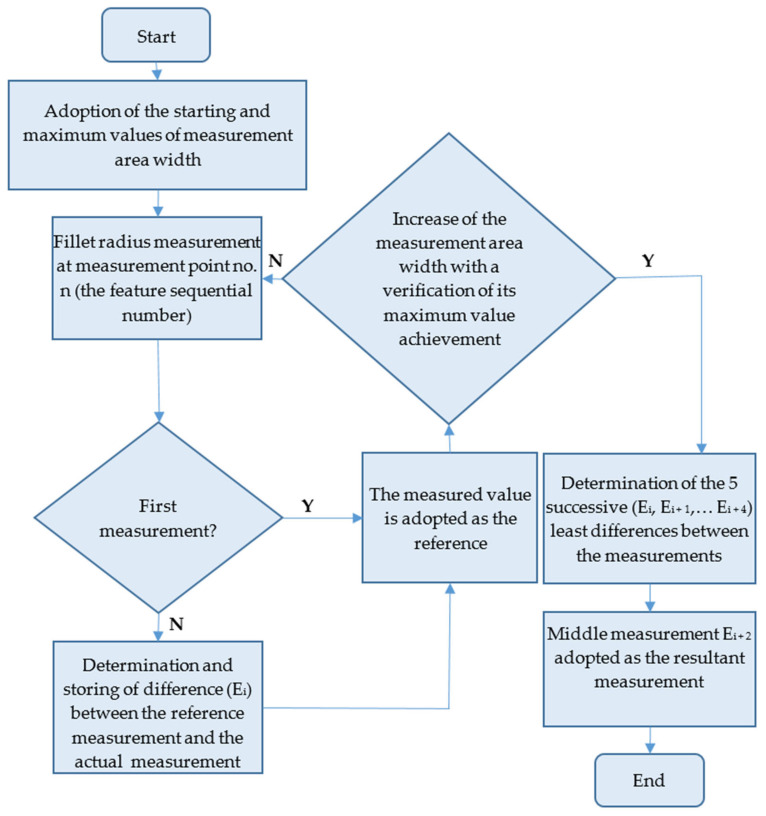
Algorithm for the fillet radius determination with varying measurement area width.

**Figure 20 sensors-22-03457-f020:**
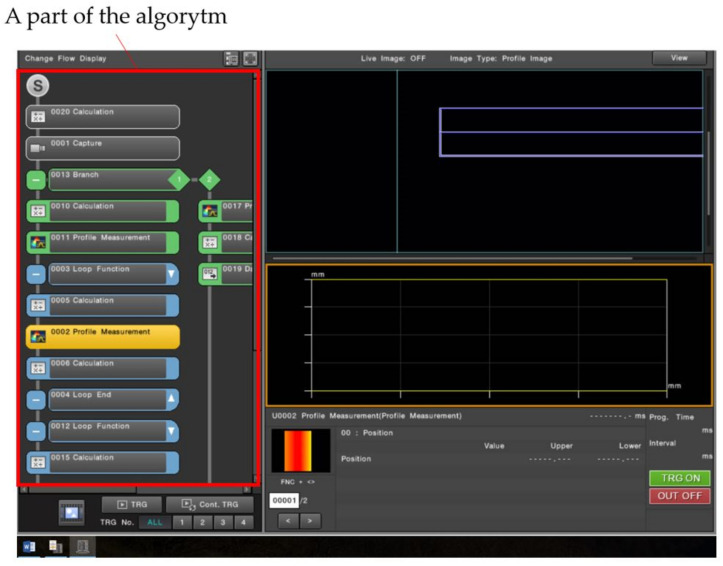
Overview of the program executed in the Keyence controller.

**Table 1 sensors-22-03457-t001:** Fillet radius R01 measurement.

Measurement Seq. No.	Imprint	Keyence	Cognex
1	1.1753	1.1745	1.1777
2	1.1793	1.1787	1.1811
3	1.1733	1.1723	1.1748
4	1.1748	1.1739	1.1764
5	1.1757	1.175	1.1782
Mean Value	1.17568	1.17488	1.17764
Standard Deviation		0.001931	0.00191

**Table 2 sensors-22-03457-t002:** Fillet radius R02 measurement.

Measurement Seq. No.	Imprint	Keyence	Cognex
1	1.4218	1.4189	1.4267
2	1.4168	1.4111	1.4205
3	1.4284	1.4232	1.4323
4	1.4321	1.4304	1.4385
5	1.4321	1.4299	1.4395
Mean Value	1.42624	1.4227	1.4315
Standard Deviation		0.006588	0.006553

**Table 3 sensors-22-03457-t003:** Fillet radius R03 measurement.

Measurement Seq. No.	Imprint	Keyence	Cognex
1	1.8886	1.8523	1.8932
2	1.8721	1.8653	1.8836
3	1.8271	1.8023	1.8324
4	1.8721	1.8485	1.8973
5	1.9046	1.8932	1.9142
Mean Value	1.8729	1.85232	1.88414
Standard Deviation		0.026945	0.025286

## Data Availability

Data is contained within the article.

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
