# Peer review of "Robotised Geometric Inspection of Thin-Walled Aerospace Casings"

_sensors, 2022, doi:10.3390/s22093457_

Round 1
Reviewer 1 Report
In my opinion, the article is more of a proposal fora measurement solution, less typical scientific work.
However, the proposed case study solution has great application
potential and may contribute to the improvement of the speed of the
quality control process .
Authors wrote that repeatability did not affected the capabilities of the inspection system because the measurement accuracy was driven by a 2D scanner and its software. So what was the accuracy of this measurement??
Authors wrote "inspection workstation included an ABB IRBP A250 2-axis positioner, on which the work- 175piece and its fixture were mounted" what about repeatability of this positioner? Did it affected the measurement?
Although some related works were mentioned in the manuscript, it would be interesting to present a comparative analysis with other methodology which could be useful. In my opinion authors sholuld mention in references for examples about articles like: "No Clamp Robotic Assembly with Use of Point Cloud Data from Low-Cost Triangulation Scanner"; "Optical scanner assisted robotic assembly"; or even "
Programming of industrial robots using the recognition of geometric signs in flexible welding process" could be useful.
best regards
Author Response
Reviewer 1
In my opinion, the article is more of a proposal fora measurement solution, less typical scientific work.
However, the proposed case study solution has great application potential and may contribute to the improvement of the speed of the quality control process .
Note as comment
Authors wrote that repeatability did not affected the capabilities of the inspection system because the measurement accuracy was driven by a 2D scanner and its software. So what was the accuracy of this measurement??
Added in text:
Each measurement is made locally, the accuracy of the measurement depends only on the accuracy of the measuring head being used, defined by the manufacturer. The manufacturer states that the Cognex DS910B provides measurement accuracy of 7 μm, while the Keyence LJ-V7060 provides measurement accuracy of 5 μm.
Authors wrote "inspection workstation included an ABB IRBP A250 2-axis positioner, on which the work- 175piece and its fixture were mounted" what about repeatability of this positioner? Did it affected the measurement?
Added in text:
The workpiece is mounted on a positioner relative to the mounting bases. The positioner's role is to position the workpiece properly for measurement. The ABB IRB 2400 robot ( with 6 degrees of freedom) on its own is not capable of providing proper head position and orientation relative to the workpiece due to the size of the measured workpiece. The repeatability of the positioner does not affect the accuracy of the measurement because the measurement is local to the indicated location and depends only on the accuracy of the measuring head used.
Although some related works were mentioned in the manuscript, it would be interesting to present a comparative analysis with other methodology which could be useful. In my opinion authors sholuld mention in references for examples about articles like: "No Clamp Robotic Assembly with Use of Point Cloud Data from Low-Cost Triangulation Scanner"; "Optical scanner assisted robotic assembly"; or even "Programming of industrial robots using the recognition of geometric signs in flexible welding process" could be useful.
Added in the text, although we do not understand the relationship between the indicated articles and the subject matter presented.
Industrial robots are used for a wide variety of operations, from simple moving materials from point A to point B to complex operations such as metalworking or assembly using sensory systems to adapt to a randomly changing environment [23,24]. Another example of adaptation algorithms or robot path correction based on camera image analysis in a welding application is included in [25].
- Suszyński M., Wojciechowski J., Żurek J. No Clamp Robotic Assembly with Use of Point Cloud Data from Low-Cost Triangulation Scanner. Tehnički vjesnik, 2018, 25(3), pp. 904-909, https://doi.org/10.17559/TV-20160922092849.
- Wojciechowski J., Suszynski M. Optical scanner assisted robotic assembly. Assembly Automation, 2017, Vol. 37 No. 4, pp. 434-441, https://doi.org/10.1108/AA-07-2016-068.
- Ciszak O., Juszkiewicz J., Suszyński M. Programming of industrial robots using the recognition of geometric signs in flexible welding process. Symmetry, 2020, 12(9), 1429, https://doi.org/10.3390/sym12091429.

Reviewer 2 Report
The chosen solution is presented only on the example of measuring the inner radius and it is not clear from the text what other types of tasks the described configuration could measure. In the introduction, the authors state that up to 1000 dimensional measurements are performed on ADT gearboxes during production. It would be good to add what types of measurements are generally performed on these gearboxes (outside the radii), what percentage of them can be performed automatically (using a robot) and what percentage of them can be measured directly as described in the article.
I would also recommend focusing on the pictures - on the flowcharts fig. 7, 10 and 17 it is necessary to unify the text size (in Fig. 7 and 17 it is too small). Figures 15 and 20 in their current form do not provide much information to the reader (the text in them is not legible), the algorithm in Figure 20 would be better presented in another form (for example, enlarge the relevant part of the image at the cost of cropping its right part).
Author Response
Reviewer 2
The chosen solution is presented only on the example of measuring the inner radius and it is not clear from the text what other types of tasks the described configuration could measure. In the introduction, the authors state that up to 1000 dimensional measurements are performed on ADT gearboxes during production. It would be good to add what types of measurements are generally performed on these gearboxes (outside the radii), what percentage of them can be performed automatically (using a robot) and what percentage of them can be measured directly as described in the article.
Added in text:
ADT's dedicated test housing has more than a thousand dimensional characteristics such as angle, radius, distance, and measurements of the relationship between bearing seats such as concentricity, flatness, etc. The proposed measurement method applies only to measurements of local features with dimensions dependent on the width of 2D scanner measurement lines. For gearboxes, these types of measurements account for 83% of the total measurements made. A number of the controlled measurements are defined as critical dimensions and in this case the measurements are mainly performed on CMM measuring centers. Of all the critical dimensions, only 82 measurements are made using a 2D scanner. The proposed solution does not eliminate all the necessary measurements, but it significantly speeds up the inspection process, this applies to the accuracy of chamfer and radius blunts as well as to the measurement of small diameters.
I would also recommend focusing on the pictures - on the flowcharts fig. 7, 10 and 17 it is necessary to unify the text size (in Fig. 7 and 17 it is too small). Figures 15 and 20 in their current form do not provide much information to the reader (the text in them is not legible), the algorithm in Figure 20 would be better presented in another form (for example, enlarge the relevant part of the image at the cost of cropping its right part).
Figures were corrected according to the reviewer's comment. Regarding Figure 20, the purpose was to prove that the proposed algorithms are implemented on a real controller. Currently the workstation is implemented for continuous production work we have no possibility at this stage to interfere and preview the solution to improve its quality. The workstation was implemented in production at Pratt&Whitney, and because part of the plant's production is military, we do not have the ability to change Figure 20 as suggested by the reviewer.

Reviewer 3 Report
The paper proposes the development of dimensional control technology for the production of accessory drive train gearbox housing. To improve the readability of the article, the following comments should be considered:
The abstract should be improved by adding the background of the problem.
The authors should add how they calibrate the measurement system.
Fig 20 presents an overview of the program executed in the Keyence controller instead of the algorithm as mentioned in line 303.
Please explain the meaning of RAPID in line 184.
Please revise Figure 7 and Figure 10 where diamond with if condition (if n>0) should be added. Please add also responses: yes, no.
In Figure 19, “N” is misplaced.
In the Paper, discussion part is missing.
The literature review is very limited, items 7-10 are authorial. Please develop the recent development in the laser measurement, geometry control...
Author Response
Reviewer 3
The paper proposes the development of dimensional control technology for the production of accessory drive train gearbox housing. To improve the readability of the article, the following comments should be considered:
The abstract should be improved by adding the background of the problem.
Added in text:
Measuring geometrical features such as full circle radii, chamfer widths, small diameters or distances with a 2D scanner is a typical application solution and does not present a problem. The main problem solved in this paper is to give an algorithm for measuring radii performed as blunts of sharp surfaces. The difficulty arises from the need to adapt the measurement field, the number of selected points and to propose a way to average the measurement. This is an approximation problem of inscribing a radius into a point-defined curve, realized under the software limitations of the Keyence and Cognex 2D scanner controllers being used.
The authors should add how they calibrate the measurement system.
Added in text:
The measurement system consisting of Kyence and Cognex 2D laser scanners are calibrated by the manufacturers and the accuracy of their measurements is defined in the manufacturer's certificate.
In this paper, the accuracy of the proposed solution was checked on a reference part approved at the gearbox manufacturer's plant (Figure 17).
Fig 20 presents an overview of the program executed in the Keyence controller instead of the algorithm as mentioned in line 303.
Regarding Figure 20, the purpose was to prove that the proposed algorithms were implemented on a real controller.
Please explain the meaning of RAPID in line 184.
Added in text:
The operating algorithm of the robotised inspection station was developed in robot ABB language RAPID and the communication was based on TCP/IP (Figure 7).
Please revise Figure 7 and Figure 10 where diamond with if condition (if n>0) should be added. Please add also responses: yes, no.
Figure 7
Figure 10
In Figure 19, “N” is misplaced.
Figure 19
In the Paper, discussion part is missing.
Reformatted as per template
The literature review is very limited, items 7-10 are authorial. Please develop the recent development in the laser measurement, geometry control...
Added in text:
Minimizing production costs implies, among other things, reducing time-consuming product inspection processes, which results in the search for alternative methods. Operations of dimensional and geometric tolerance control have so far often been performed on coordinate measuring machines (CMMs). Contact measurement with a CMM has well-defined accuracy and measurement uncertainty [3]. A cheaper alternative are non-contact measurements, unfortunately the measurement uncertainty of such solutions is at least one order of magnitude higher [4]. Non-contact measurements, however, have an advantage in the speed of measurement data acquisition, allowing with a higher density of points in a rapid acquisition of cloud points. Therefore, they have been widely used in reverse engineering [5, 6]. In the last few years, a great effort has been made to improve the accuracy of laser systems [7], and their use in inspection activities is increasing every day. The most common laser systems used in metrology applications are those based on laser triangulation using a laser beam (ruler). The reason for the popularity of 2D scanning technology is the relatively low price, high precision, relatively low computational cost compared to methods based on structured light, holography or image analysis. Interesting information can be found in the work [8] where two different scanning systems were analyzed: a laser triangulation sensor and a contact probe, both mounted on a coordinate measuring machine, and the results were analyzed to compare them in terms of geometric and dimensional tolerances.
- Li Y., Gu P. Free-form surface inspection techniques state of the art review. Computer-Aided Design, 2004, 36(13), pp. 1395-1417, https://doi.org/10.1016/j.cad.2004.02.009.
- Feng H. Y., Liu Y., Xi F. Analysis of digitizing errors of a laser scanning system. Precision Engineering, 2001, 25(3), pp. 185-191, https://doi.org/10.1016/S0141-6359(00)00071-4.
- Son S., Park H., Lee K. H. Automated laser scanning system for reverse engineering and inspection. International Journal of machine tools and manufacture, 2002, 42(8), pp. 889-897, https://doi.org/10.1016/S0890-6955(02)00030-5.
- Carbone V., Carocci M., Savio E., Sansoni G., De Chiffre L. Combination of a vision system and a coordinate measuring machine for the reverse engineering of freeform surfaces. The International Journal of Advanced Manufacturing Technology, 2001, 17(4), pp. 263-271, https://doi.org/10.1007/s001700170179.
- Cuesta E., Rico J. C., Fernández P., Blanco D., Valiño G. Influence of roughness on surface scanning by means of a laser stripe system. The International Journal of Advanced Manufacturing Technology, 2009, 43(11), pp. 1157-1166, https://doi.org/10.1007/s00170-008-1794-9.
- Martínez S., Cuesta E., Barreiro J., Álvarez B. Analysis of laser scanning and strategies for dimensional and geometrical control. The International Journal of Advanced Manufacturing Technology, 2010, 46(5), pp. 621-629, https://doi.org/10.1007/s00170-009-2106-8.

Reviewer 4 Report
I cannot suggest the publishing of the manuscript in this form, because the article is not fulfilling the scientific requirements of a high quality IF/Q1 journal like the “Sensors”.
In my opinion a number of important issues should be addressed through a major revision, which improve the present manuscript.
My comments are the following:
1.) The structure of the study is not fulfilling the formal requirements of the journal (Introduction, Materials and Methods, Results, Discussion, Conclusions). Please check the template.
2.) Furthermore, the required parts of the section Introduction are missing: “… Finally, briefly mention the main aim of the work and highlight the main conclusions.” Please check the template.
3.) The significance and the novelty of the study is not well defined.
4.) The scientific background of the article is missing. The current state of the research field is not reviewed carefully; the relevant publications are missing.
5.) The scientific main added values of the study is also missing which are basic requirements of a high quality journal. Please highlight the main contribution of the research.
Author Response
Reviewer 4
1.) The structure of the study is not fulfilling the formal requirements of the journal (Introduction, Materials and Methods, Results, Discussion, Conclusions). Please check the template.
Reformatted as per template
2.) Furthermore, the required parts of the section Introduction are missing: “… Finally, briefly mention the main aim of the work and highlight the main conclusions.” Please check the template.
Added from line 43
Minimizing production costs implies, among other things, reducing time-consuming product inspection processes, which results in the search for alternative methods. Operations of dimensional and geometric tolerance control have so far often been performed on coordinate measuring machines (CMMs). Contact measurement with a CMM has well-defined accuracy and measurement uncertainty [3]. A cheaper alternative are non-contact measurements, unfortunately the measurement uncertainty of such solutions is at least one order of magnitude higher [4]. Non-contact measurements, however, have an advantage in the speed of measurement data acquisition, allowing with a higher density of points in a rapid acquisition of cloud points. Therefore, they have been widely used in reverse engineering [5, 6]. In the last few years, a great effort has been made to improve the accuracy of laser systems [7], and their use in inspection activities is increasing every day. The most common laser systems used in metrology applications are those based on laser triangulation using a laser beam (ruler). The reason for the popularity of 2D scanning technology is the relatively low price, high precision, relatively low computational cost compared to methods based on structured light, holography or image analysis. Interesting information can be found in the work [8] where two different scanning systems were analyzed: a laser triangulation sensor and a contact probe, both mounted on a coordinate measuring machine, and the results were analyzed to compare them in terms of geometric and dimensional tolerances.
3.) The significance and the novelty of the study is not well defined.
Added from line 62
Measuring geometrical features such as full circle radii, chamfer widths, small diameters or distances with a 2D scanner is a typical application solution and does not present a problem. The main problem solved in this paper is to give an algorithm for measuring radii performed as blunts of sharp surfaces. The difficulty arises from the need to adapt the measurement field, the number of selected points and to propose a way to average the measurement. This is an approximation problem of inscribing a radius into a point-defined curve, realized under the software limitations of the Keyence and Cognex 2D scanner controllers being used.
4.) The scientific background of the article is missing. The current state of the research field is not reviewed carefully; the relevant publications are missing.
Added [3-8,23-25] as well as descriptions and references in the text.
- Li Y., Gu P. Free-form surface inspection techniques state of the art review. Computer-Aided Design, 2004, 36(13), pp. 1395-1417, https://doi.org/10.1016/j.cad.2004.02.009.
- Feng H. Y., Liu Y., Xi F. Analysis of digitizing errors of a laser scanning system. Precision Engineering, 2001, 25(3), pp. 185-191, https://doi.org/10.1016/S0141-6359(00)00071-4.
- Son S., Park H., Lee K. H. Automated laser scanning system for reverse engineering and inspection. International Journal of machine tools and manufacture, 2002, 42(8), pp. 889-897, https://doi.org/10.1016/S0890-6955(02)00030-5.
- Carbone V., Carocci M., Savio E., Sansoni G., De Chiffre L. Combination of a vision system and a coordinate measuring machine for the reverse engineering of freeform surfaces. The International Journal of Advanced Manufacturing Technology, 2001, 17(4), pp. 263-271, https://doi.org/10.1007/s001700170179.
- Cuesta E., Rico J. C., Fernández P., Blanco D., Valiño G. Influence of roughness on surface scanning by means of a laser stripe system. The International Journal of Advanced Manufacturing Technology, 2009, 43(11), pp. 1157-1166, https://doi.org/10.1007/s00170-008-1794-9.
- Martínez S., Cuesta E., Barreiro J., Álvarez B. Analysis of laser scanning and strategies for dimensional and geometrical control. The International Journal of Advanced Manufacturing Technology, 2010, 46(5), pp. 621-629, https://doi.org/10.1007/s00170-009-2106-8.
- Suszyński M., Wojciechowski J., Żurek J. No Clamp Robotic Assembly with Use of Point Cloud Data from Low-Cost Triangulation Scanner. Tehnički vjesnik, 2018, 25(3), pp. 904-909, https://doi.org/10.17559/TV-20160922092849.
- Wojciechowski J. Suszynski M. Optical scanner assisted robotic assembly. Assembly Automation, 2017, Vol. 37 No. 4, pp. 434-441, https://doi.org/10.1108/AA-07-2016-068.
- Ciszak O., Juszkiewicz J., Suszyński M. Programming of industrial robots using the recognition of geometric signs in flexible welding process. Symmetry, 2020, 12(9), 1429, https://doi.org/10.3390/sym12091429.
5.) The scientific main added values of the study is also missing which are basic requirements of a high quality journal. Please highlight the main contribution of the research.
Added from line 389
The main contribution of the research is the development of the algorithm and its implementation in the measuring device. This algorithm is designed to increase or decrease the width of the measuring field depending on the tested radius (0.4 to 1.8 mm) and to determine its different values automatically.

Round 2
Reviewer 4 Report
I accept the article and support the publication of it.